# The Technical–Tactical Patterns of Scoring Actions in Male Visually Impaired Judokas: A Weight Category Analysis

Alfonso Gutiérrez-Santiago [1], Anxo Fernández-Moscoso [1], Xoana Reguera-López-de-la-Osa [2,*],
Antonio José Silva-Pinto [1], Juan Carlos Argibay-González [1] and Iván Prieto-Lage [1,*]

1   Observational Research Group, Faculty of Education and Sport, University of Vigo, 36005 Pontevedra, Spain;
    ags@uvigo.es (A.G.-S.); anxelofm@gmail.com (A.F.-M.); toze-pinto@hotmail.com (A.J.S.-P.);
    juan.carlos.argibay@uvigo.gal (J.C.A.-G.)
2   Education, Physical Activity and Health Research Group (Gies10-DE3), Galicia Sur Health Research,
    Institute (IIS Galicia Sur), SERGAS-UVIGO, 36208 Vigo, Spain
*   Correspondence: xreguera@uvigo.es (X.R.-L.-d.-l.-O.); ivanprieto@uvigo.es (I.P.-L.)

**Abstract:** In Para-judo, the technical actions and tactical situations suitable for scoring in a combat have not been studied. The aim of this research was to determine the technical–tactical patterns of scoring actions in Para-judo, focusing on weight categories. An observational methodology was employed to conduct this study. The sample consisted of all male judokas with visual impairment (*n* = 172) in the senior category who participated in the World Championships held in Odivelas in 2018. A total of 232 combats and 313 scoring actions were analyzed. To obtain the results, we used different analytical techniques with SPSS: descriptive analysis, chi-square test, and T-pattern analysis. The significance level used was $p < 0.05$. Key findings show that the majority of scoring actions occurred in the first two minutes of combat, highlighting the importance of early initiative. Techniques such as ashi-waza, te-waza, and sutemi-waza demonstrated particular effectiveness across categories, with a focus on movements like sumi-otoshi, ouchi-gari, and ko-soto-gake. Moreover, the most common grip—lapel-sleeve—proved advantageous, aiding judokas in controlling the bout and achieving scores, especially when coupled with effective transition techniques. The study's weight-specific analysis further revealed distinct patterns, emphasizing the importance of tailored training approaches. For instance, lighter judokas benefited from countering leg attacks with arm or sacrifice techniques to score waza-ari, while heavier judokas favored leg-to-leg counterattacks, often achieving ippon. The results suggest that judokas and coaches could benefit from a more systematic focus on grip stability, strategic positional work, and transition efficiency, particularly from standing to ground. This research contributes valuable insights into optimal techniques and strategies for visually impaired judokas, providing clear guidelines for training and competition.

**Keywords:** para-judo; technique; scoring; visually impaired; T-pattern; weight categories

## 1. Introduction

During a judo match, scoring is the determining criterion for establishing the winner. This underscores the importance of understanding the specific actions that lead to successful scoring. Such knowledge is essential for optimizing the performance of judokas, particularly those with visual impairments (VI), thereby maximizing their opportunities for competitive success. The observation and analysis of technical–tactical actions that lead to scoring have improved significantly in recent years, partly due to advances in recording, tracking, and data coding technologies [1]. These advancements allow not only for the description and prediction of performance but also for intervention in the factors that determine sporting victory [2], which is particularly valuable when applied to the movements and specific patterns that result in scoring actions.

Judo is a discipline in constant evolution, where the rules are updated periodically to make the sport more visually appealing, promoting a more offensive style of combat and

penalizing actions that disrupt the flow of the match [3,4]. As a result, the most commonly used techniques have changed, favoring a more dynamic approach with an emphasis on quick and effective throws [5]. In parallel, these adaptations have also been integrated into Para-judo, ensuring that both modalities evolve in unison. Consequently, the analysis of effective techniques that lead to scoring must consider the particular context of Para-judo. It is important to note that para-judokas start the match with a grip; when they release the grip, the match stops. There is less struggle for the grip, and the resting times are longer because they take more time to return to the starting position after a matte due to their visual impairment. It is essential to consider the specific adaptations, both tactical and technical, that arise from the unique characteristics of these athletes.

Despite the constant updates to the rules, the primary objective of judo remains the same: to achieve an ippon to secure victory. The pursuit of strategies to achieve this goal has been a central focus of judo research [6], as it allows for the identification of the most effective sequences and techniques. Although individual techniques have evolved over the years, the groups of techniques most frequently used have remained relatively stable. Ashi-waza (leg techniques) are the most common, followed by te-waza (arm techniques), sutemi-waza (sacrifice techniques), and koshi-waza (hip techniques) [7,8]. This consistency in the use of certain technical groups suggests the existence of effective scoring patterns, which may also be present in Para-judo.

Success in judo does not depend solely on the technique used. Other factors also play a role as prerequisites for a successful throw. Among these factors is the grip (kumikata). The sleeve-lapel grip is the most common in high-level competitions [9]. This type of grip provides optimal control of the opponent, which is essential for preparing any offensive action. Additionally, prior movements are crucial for the effectiveness of the techniques. There are more effective attack directions to score [10,11]. It has been documented that the most successful judokas employ at least four different movements before attacking [12]. This suggests a complex strategy to prepare and destabilize the opponent. Attacking in multiple directions, at least three, increases unpredictability and makes it difficult for the opponent to defend. This enhances the chances of success [13,14]. High-level judokas tend to vary their attack directions to create more points of imbalance and overcome defenses [14,15]. Furthermore, studies suggest that the ability to execute techniques bilaterally is associated with better combat performance and greater efficiency in technique application [16].

The duration of the combats also provides relevant information to understand when most scoring actions occur. In the combats of judokas with visual impairment at the 2018 IBSA World Judo Championships, 32.8% of the combats ended in the first minute, 25.4% in the second minute, 16.4% in the third minute, 18.5% in the fourth minute, and only 6.9% reached the Golden Score period [17]. These data suggest a trend toward the early resolution of combats. This has implications for both physical preparation and tactical strategy.

Regarding the scores achieved, comparative studies between sighted judokas and para-judokas have shown similar percentages for each type of scoring: ippon (50–60% for sighted judokas and 54–63% for para-judokas), waza-ari (19–25% versus 16–17%), and yuko (38–47% versus 41–47%) [18]. The observed differences between both groups reveal that Olympic judokas achieve more waza-ari scores, while para-judokas show a higher percentage of ippon. While para-judokas demonstrate statistically superior performance in terms of ippon, it is important to recognize that their context and combat strategies differ. They tend to execute cleaner and more effective techniques, leading to successful projections. This results in a higher number of ippon, but does not necessarily imply that they perform techniques more efficiently than sighted judokas.

However, these studies have a limitation. The sample included judokas from various weight categories, but the analysis did not differentiate among them. It has been shown that judokas from different weight categories exhibit distinct performance characteristics [19]. Therefore, drawing conclusions from a sample that combines multiple weight categories to apply to a specific one can be misleading [20].

In this context, the present study focuses on identifying patterns of technical–tactical actions that result in scoring in judokas with visual impairment, differentiated by weight categories. Unlike most previous research, which has explored aspects such as combat duration [17] or the legitimacy of the Para-judo classification [21,22], this work aims to contribute to the field with a direct focus on the patterns that lead to successful actions, providing valuable information for the training planning of judokas with visual impairment. Thus, the specific objective of this study is to determine which technical–tactical actions enable judokas with visual impairment to score. This will allow coaches and technical staff to develop evidence-based training strategies and improve the competitive performance of these athletes.

## 2. Methods

### 2.1. Design

To conduct this research, we employed an observational methodology [23], which provides the necessary rigor and flexibility to study the episodes that occur naturally in judo combat. The observational design used [24] was nomothetic (we analyzed the effectiveness of scoring actions across all participants), longitudinal (we assessed whether there is stability in the behavior observed across different combats), and multidimensional (the dimensions correspond to the criteria of the observation instrument). A series of decisions arise from this design, which will be presented below.

### 2.2. Sample

The sample for this study consisted of all male judokas with visual impairment ($n$ = 172) in the senior category who participated in the World Championship held in Odivelas (Portugal) in 2018. A total of 232 combats were observed. The International Blind Sports Federation (IBSA) granted us the necessary permissions to conduct the research. Furthermore, the study was approved by the Ethics Committee of the Faculty of Education and Sports Sciences at the University of Vigo (Application 01/1019).

### 2.3. Instruments

To conduct this study, we based our work on the Score Action-Judo (SA-JUDO) observation instrument, used in previous research [25]. We modified the criterion "Ground Actions" by adding a new category called "Direct Action", resulting in this new version of the instrument: SA-JUDO v.2. Like its predecessor, this new version combines a field format with a category system. SA-JUDO v.2 encompasses a set of criteria that will allow us to determine the technical and tactical characteristics of scoring actions in judo combat, not only for sighted judokas but also for those with visual impairments.

SA-JUDO v.2 conforms to the proposed observational design, being multidimensional in nature and consisting of the structure of criteria and categories presented in Table 1. Each of these dimensions results in an observational instrument that meets the conditions of exhaustiveness and mutual exclusivity (E/ME). All scoring actions were coded and recorded using LINCE software v.1.4 [26].

**Table 1.** Observational instrument.

| Criterion | Code | Description | Code | Description |
|---|---|---|---|---|
| Time | 1M | 1st min: $0'00''$–$1'00''$ | 3M | 3rd min: $2'01''$–$3'00''$ |
| | 2M | 2nd min: $1'01''$–$2'00''$ | 4M | 4th min: $3'01''$–$4'00''$ |
| | GS | Golden Score: extra time—tiebreaker | | |
| Partial Score | WIN | The judoka that achieves the scoring actions is winning | | |
| | EVEN | The judoka that achieves the scoring actions has an even score | | |
| | LOSE | The judoka that achieves the scoring actions is losing | | |

**Table 1.** *Cont.*

| Criterion | Code | Description | Code | Description |
|---|---|---|---|---|
| Penalties | S1 | When a scoring action is achieved the judoka has a shido | | |
| | S2 | When a scoring action is achieved the judoka has two shido | | |
| | NS | When a scoring action is achieved, the judoka has not a shido. | | |
| Fighting Situation | UPF | The scoring action is achieved standing up | | |
| | GRF | The scoring action is achieved on the ground | | |
| Judoka | SJ | Scoring judoka | NSJ | Non-scoring judoka |
| Grip | LS | Lapel-Sleeve | SBA | Sleeve-Back |
| | SS | Sleeve-Sleeve | SBE | Sleeve-Belt |
| | LL | Lapel-Lapel | LBA | Lapel-Back |
| | LBE | Lapel-Belt | CG | Crossed Grip |
| | BH | Bear Hug | | |
| | S | Grip performed with a hand over the sleeve of the opponent | | |
| | L | Grip performed with a single hand over the lapel of the opponent | | |
| | B | Grip performed with a single hand over the back of the opponent | | |
| Movement | FW | Direction of the movement prior to the scoring action: Forward | | |
| | FWR | Idem: Forward Right | BWL | Idem: Backwards left |
| | FWL | Idem: Forward Left | R | Idem: Right |
| | BW | Idem: Backwards | LFT | Idem: Left |
| | BWR | Idem: Backwards Right | TS | Tai Sabaki |
| | ST | Static | | |
| Katame waza situation | DP | Dominant position | PP | Prone position |
| | INP | Inferior position | SP | Supine position |
| | RMP | Reverse mount position | MSD | Mounted in the same direction |
| | 4P | All-fours position | LPO | Lateral position |
| | LTR | Leg Trap | BL | Between the legs |
| | LIN | Legs intertwined | TL | Trapped leg |
| | SIDE | The judoka is on their side | | |
| Ground Actions | DAC | Direct action | RAC | Rolling action |
| | GPA | Guard pass action | REL | Removing the entangled leg |
| | TTEC | Transition technique | | |
| Score | IP | Ippon | W | Waza Ari |
| Grouped techniques | TEWZ | Te-Waza | OSAEWZ | Osaekomi-Waza |
| | KOSHIWZ | Koshi-Waza | SHIMEWZ | Shime-Waza |
| | ASHIWZ | Ashi-Waza | KANSETWZ | Kansetsu-Waza |
| | SUTWZ | Sutemi-Waza | | |
| Individual techniques | The techniques used in this instrument number over one hundred. They are included in the official Kodokan classification created by Jigoro Kano [27]. | | | |

*2.4. Procedure*

The videos analyzed in this study were recorded directly at the venue where the competition took place, using three Sony HDR-PJ410 cameras. Following the recommendations of previous research [28], each camera was used to record videos in a specific combat area, ensuring ecological validity.

Following the modification of the observation instrument and the creation of the new version SA-JUDO v.2, the validity of its construct was assessed by verifying its consistency with the theoretical framework. Additionally, two experts in observational methodology and judo were consulted, both of whom demonstrated 100% agreement with the instrument. Both experts had extensive experience in observational methodology (construction of different observation instruments in various sports disciplines, especially in combat sports

and racket sports) and in judo (both were 1st DAN black belts in judo). For this purpose, the two experts answered a questionnaire on the observation instrument, analyzing its suitability for the reality of the competition and following the same procedure as previous studies [29]. The two experts were provided with a comprehensive description of the observation instrument, the objects of the investigation, and instructions for answering the questionnaire. The questionnaire consisted of five items (with a Likert scale of five levels) about its suitability to the object of study, compliance with the criteria of completeness and mutual exclusivity, clarity in the wording of the categories, and the degree of objectivity that allows the data collection to be unified by various observers.

Once the videos of the different matches were obtained, a combined video was created using Wondershare Filmora v.10.0.2.1. In this video, confrontations that included the scoring actions of the judokas were selected and organized in chronological order. This process resulted in a more manageable file for the recording instrument (LINCE).

After appropriate training in the use of the recording instrument and the observational instrument SA-JUDO v.2, the quality of the data to be recorded was evaluated by two expert observers. To ensure rigor in the coding process [30], the quality of the recorded data was controlled by calculating intra- and inter-observer agreement using Cohen's Kappa coefficient [31], calculated through the LINCE software. Intra-observer agreement was initially assessed on one-third of the actions, yielding a Kappa value of 0.90 for Observer 1 and 0.95 for Observer 2. Subsequently, inter-observer agreement was calculated for all techniques, yielding a Kappa value of 0.92.

After editing the video, creating the observational instrument, training the observers, and ensuring their quality, Observer 2 was responsible for observing and recording the data from the videos using LINCE v.1.4.

After the recording, we obtained an Excel file containing the sequential order of all registered behavior codes, along with their timing and duration expressed in frames. The versatility of this file allowed us to perform successive transformations for various analyses [25].

*2.5. Data Analysis*

All statistical analyses were conducted using the IBM Statistical Package for the Social Sciences, version 25.0 (IBM-SPSS Inc., Chicago, IL, USA). The relationship between the different categories of this study was calculated using the chi-square test of independence ($\chi^2$). Statistical significance was assumed for $p < 0.05$. Additionally, the effect size was calculated using Cramér's V to evaluate the strength of the observed associations, with the following interpretation: 0.00–0.10 (very weak), 0.10–0.20 (weak), 0.20–0.30 (moderate), 0.30–0.40 (relatively strong), 0.40–0.50 (strong), and 0.50 or more (very strong). An analysis of the adjusted residuals was also performed to highlight significant deviations from the expected frequencies, providing additional information on the relationships between the variables.

To analyze the patterns, T-Patterns were detected using Theme v.5.0 [32] with a significance level of 0.005, which implies that the probability of accepting a critical interval due to chance is 0.5%. Initially, a minimum number of occurrences of five was set, followed by three, without discarding patterns with occurrences equal to or greater than five and three. Additionally, redundancy reduction was activated at 90% to avoid identifying similar T-Patterns. This software reveals hidden structures and unobservable aspects of sports techniques [33], making its application highly effective in sports sciences [32]. The graphical representation is a dendrogram that illustrates the behaviors or actions under study, highlighting the connections between the different technical–tactical aspects of the scoring actions. The dendrogram consists of two clearly differentiated parts. The left quadrant represents the established relationship between the different categories that make up the technical and tactical actions of the combat and should be read sequentially, like a tree diagram, from top to bottom. The right quadrant indicates how frequently these relationships occur, represented by lines extending from the top to the bottom.

## 3. Results

### 3.1. Descriptive Analysis of the Sample

Table 2 presents the descriptive analysis of the studied sample. The 172 judokas with visual impairment in the sample participated in a total of 232 combats. Of these, 211 combats featured at least one scoring action, while 21 did not. The reasons for this included the accumulation of three shido (without any scoring actions occurring during the combat) or injury to a competitor, who had to withdraw from the combat. In total, 313 scoring actions were recorded: 129 ippon and 184 waza-ari. Among the 313 scoring actions, 261 occurred while standing, and 52 took place on the ground.

**Table 2.** Description of the sample by weight categories.

| | −60 kg | −66 kg | −73 kg | −81 kg | −90 kg | −100 kg | +100 kg | Total |
|---|---|---|---|---|---|---|---|---|
| Competitors | 29 | 30 | 31 | 25 | 21 | 19 | 17 | 172 |
| Combats | 38 | 40 | 41 | 34 | 29 | 27 | 23 | 232 |
| Combats without SA | 2 [1] | 2 [1] | 2 [2] | 4 [2] | 3 [1] | 5 [1] | 3 [1] | 21 |
| Combats with SA | 36 | 38 | 39 | 30 | 26 | 22 | 20 | 211 |
| Score actions | 53 (20 I y 33 W) | 54 (21 I y 33 W) | 66 (19 I y 47 W) | 51 (19 I y 32 W) | 33 (20 I y 13 W) | 27 (17 I y 10 W) | 29 (13 I y 16 W) | 313 (129 I y 184 W) |

Note: SA = Score action; I = Ippon; W = Waza-ari. [1] In all the combats, there were no scoring actions; the winner was determined by the opponent accumulating three shido penalties. [2] In all the combats, there were no scoring actions: in one combat, the winner was due to the opponent's injury, while in the others, it was due to the opponent accumulating three shido penalties.

### 3.2. Descriptive Analysis of Scoring Actions

Table 3 presents the descriptive analysis of standing scoring actions in judo (*n* = 261).

**Table 3.** Frequency, %, and chi-square of categories related to scoring actions in standing judo.

| Variables | Total | | −60 kg | | −66 kg | | −73 kg | | −81 kg | | −90 kg | | −100 kg | | +100 kg | | Chi-Square | |
|---|---|---|---|---|---|---|---|---|---|---|---|---|---|---|---|---|---|---|
| | Fr. | % | Fr. | % | Fr. | % | Fr. | % | Fr. | % | Fr. | % | Fr. | % | Fr. | % | $\chi^2$ | $p$ |
| Time | | | | | | | | | | | | | | | | | 19.468 | 0.727 |
| 1st Minute | 124 | 47.5 | 20 | 43.5 | 17 | 37 | 28 | 48.3 | 23 | 52.3 | 12 | 42.9 | 13 | 68.4 | 11 | 55 | | |
| 2nd Minute | 73 | 28 | 12 | 26.1 | 14 | 30.4 | 14 | 24.1 | 12 | 27.3 | 9 | 32.1 | 4 | 21.1 | 8 | 40 | | |
| 3rd Minute | 36 | 13.8 | 7 | 15.2 | 8 | 17.4 | 8 | 13.8 | 7 | 15.9 | 4 | 14.3 | 1 | 5.3 | 1 | 5 | | |
| 4th Minute | 23 | 8.8 | 5 | 10.9 | 7 | 15.2 | 6 | 10.3 | 1 | 2.3 | 3 | 10.7 | 1 | 5.3 | 0 | 0 | | |
| Golden Score | 5 | 1.9 | 2 | 4.3 | 0 | 0 | 2 | 3.4 | 1 | 2.3 | 0 | 0 | 0 | 0 | 0 | 0 | | |
| Partial Score | | | | | | | | | | | | | | | | | 14.295 | 0.282 |
| Winning | 47 | 18 | 6 | 13 | 10 | 21.7 | 16 | 27.6 | 9 | 20.5 | 5 | 17.9 | 1 | 5.3 | 0 | 0 | | |
| Even score | 196 | 75.1 | 35 | 76.1 | 34 | 73.9 | 38 | 65.5 | 31 | 70.5 | 22 | 78.6 | 17 | 89.5 | 19 | 95 | | |
| Losing | 18 | 6.9 | 5 | 10.9 | 2 | 4.3 | 4 | 6.9 | 4 | 9.1 | 1 | 3.6 | 1 | 5.3 | 1 | 5 | | |
| SJ Penalty | | | | | | | | | | | | | | | | | 11.196 | 0.512 |
| Shido1 | 56 | 21.5 | 11 | 23.9 | 9 | 19.6 | 13 | 22.4 | 9 | 20.5 | 7 | 25 | 1 | 5.3 | 6 | 30 | | |
| Shido2 | 22 | 8.4 | 7 | 15.2 | 3 | 6.5 | 7 | 12.1 | 2 | 4.5 | 1 | 3.6 | 1 | 5.3 | 1 | 5 | | |
| Without Shido | 183 | 70.1 | 28 | 60.9 | 34 | 73.9 | 38 | 65.5 | 33 | 75 | 20 | 71.4 | 17 | 89.5 | 13 | 65 | | |
| NSJ Penalty | | | | | | | | | | | | | | | | | 9.59 | 0.652 |
| Shido1 | 63 | 24.1 | 8 | 17.4 | 12 | 26.1 | 13 | 22.4 | 10 | 22.7 | 11 | 39.3 | 4 | 21.1 | 5 | 25 | | |
| Shido2 | 27 | 10.3 | 7 | 15.2 | 6 | 13 | 6 | 10.3 | 3 | 6.8 | 3 | 10.7 | 0 | 0 | 2 | 10 | | |
| Without Shido | 171 | 65.5 | 31 | 67.4 | 28 | 60.9 | 39 | 67.2 | 31 | 70.5 | 14 | 50 | 15 | 78.9 | 13 | 65 | | |
| SJ Grip | | | | | | | | | | | | | | | | | 74.541 | 0.098 |
| Lapel-Sleeve | 159 | 60.9 | 33 | 71.7 | 26 | 56.5 | 34 | 58.6 | 28 | 63.6 | 15 | 53.6 | 10 | 52.6 | 13 | 65 | | |
| Sleeve-Sleeve | 8 | 3.1 | 2 | 4.3 | 3 | 6.5 | 0 | 0 | 1 | 2.3 | 1 | 3.6 | 1 | 5.3 | 0 | 0 | | |
| Lapel-Lapel | 3 | 1.1 | 0 | 0 | 0 | 0 | 0 | 0 | 0 | 0 | 1 | 3.6 | 0 | 0 | 2 | 10 | | |
| Sleeve-Back | 40 | 15.3 | 4 | 8.7 | 7 | 15.2 | 10 | 17.2 | 7 | 15.9 | 5 | 17.9 | 4 | 21.1 | 3 | 15 | | |
| Crossed Grip | 5 | 1.9 | 1 | 2.2 | 4 | 8.7 | 0 | 0 | 0 | 0 | 0 | 0 | 0 | 0 | 0 | 0 | | |
| Bear Hug | 6 | 2.3 | 1 | 2.2 | 2 | 4.3 | 1 | 1.7 | 1 | 2.3 | 1 | 3.6 | 0 | 0 | 0 | 0 | | |
| Sleeve only | 3 | 1.1 | 1 | 2.2 | 0 | 0 | 1 | 1.7 | 1 | 2.3 | 0 | 0 | 0 | 0 | 0 | 0 | | |
| Lapel only | 16 | 6.1 | 1 | 2.2 | 1 | 2.2 | 5 | 8.6 | 2 | 4.5 | 5 | 17.9 | 2 | 10.5 | 0 | 0 | | |
| Back only | 0 | 0 | 0 | 0 | 0 | 0 | 0 | 0 | 0 | 0 | 0 | 0 | 0 | 0 | 0 | 0 | | |
| Belt only | 4 | 1.5 | 2 | 4.3 | 0 | 0 | 0 | 0 | 1 | 2.3 | 0 | 0 | 0 | 0 | 1 | 5 | | |
| Lapel-Back | 10 | 3.8 | 0 | 0 | 1 | 2.2 | 5 | 8.6 | 1 | 2.3 | 0 | 0 | 2 | 10.5 | 1 | 5 | | |
| Sleeve-Belt | 7 | 2.7 | 1 | 2.2 | 2 | 4.3 | 2 | 3.4 | 2 | 4.5 | 0 | 0 | 0 | 0 | 0 | 0 | | |
| NSJ Grip | | | | | | | | | | | | | | | | | 60.953 | 0.441 |
| Lapel-Sleeve | 206 | 78.9 | 38 | 82.6 | 37 | 80.4 | 48 | 82.8 | 35 | 79.5 | 18 | 64.3 | 14 | 73.7 | 16 | 80 | | |
| Sleeve-Sleeve | 2 | 0.8 | 0 | 0 | 0 | 0 | 0 | 0 | 1 | 2.3 | 1 | 3.6 | 0 | 0 | 0 | 0 | | |
| Lapel-Lapel | 6 | 2.3 | 3 | 6.5 | 0 | 0 | 0 | 0 | 0 | 0 | 1 | 3.6 | 1 | 5.3 | 1 | 5 | | |
| Sleeve-Back | 18 | 6.9 | 3 | 6.5 | 3 | 6.5 | 2 | 3.4 | 4 | 9.1 | 2 | 7.1 | 1 | 5.3 | 3 | 15 | | |
| Crossed Grip | 4 | 1.5 | 0 | 0 | 3 | 6.5 | 0 | 0 | 0 | 0 | 0 | 0 | 1 | 5.3 | 0 | 0 | | |
| Bear Hug | 1 | 0.4 | 1 | 2.2 | 0 | 0 | 0 | 0 | 0 | 0 | 0 | 0 | 0 | 0 | 0 | 0 | | |
| Sleeve only | 8 | 3.1 | 0 | 0 | 1 | 2.2 | 3 | 5.2 | 2 | 4.5 | 2 | 7.1 | 0 | 0 | 0 | 0 | | |
| Lapel only | 13 | 5 | 1 | 2.2 | 1 | 2.2 | 3 | 5.2 | 2 | 4.5 | 4 | 14.3 | 2 | 10.5 | 0 | 0 | | |
| Back only | 1 | 0.4 | 0 | 0 | 0 | 0 | 1 | 1.7 | 0 | 0 | 0 | 0 | 0 | 0 | 0 | 0 | | |

**Table 3.** *Cont.*

| Variables | Total | | −60 kg | | −66 kg | | −73 kg | | −81 kg | | −90 kg | | −100 kg | | +100 kg | | Chi-Square | |
|---|---|---|---|---|---|---|---|---|---|---|---|---|---|---|---|---|---|---|
| | Fr. | % | Fr. | % | Fr. | % | Fr. | % | Fr. | % | Fr. | % | Fr. | % | Fr. | % | $\chi^2$ | $p$ |
| Belt only | 1 | 0.4 | 0 | 0 | 1 | 2.2 | 0 | 0 | 0 | 0 | 0 | 0 | 0 | 0 | 0 | 0 | | |
| Lapel-Back | 1 | 0.4 | 0 | 0 | 0 | 0 | 1 | 1.7 | 0 | 0 | 0 | 0 | 0 | 0 | 0 | 0 | | |
| Sleeve-Belt | 0 | 0 | 0 | 0 | 0 | 0 | 0 | 0 | 0 | 0 | 0 | 0 | 0 | 0 | 0 | 0 | | |
| SJ Movement | | | | | | | | | | | | | | | | | 75.939 | 0.026 |
| Forward | 16 | 6.1 | 2 | 4.3 | 2 | 4.3 | 4 | 6.9 | 2 | 4.5 | 2 | 7.1 | 4 | 21.1 | 0 | 0 | | |
| Forward Right | 4 | 1.5 | 2 | 4.3 | 0 | 0 | 1 | 1.7 | 0 | 0 | 0 | 0 | 1 | 5.3 | 0 | 0 | | |
| Forward Left | 7 | 2.7 | 1 | 2.2 | 1 | 2.2 | 1 | 1.7 | 1 | 2.3 | 0 | 0 | 0 | 0 | 3 | 15 | | |
| Backwards | 16 | 6.1 | 6 | 13 | 2 | 4.3 | 4 | 6.9 | 1 | 2.3 | 1 | 3.6 | 1 | 5.3 | 1 | 5 | | |
| Backwards Right | 7 | 2.7 | 0 | 0 | 2 | 4.3 | 3 | 5.2 | 0 | 0 | 0 | 0 | 0 | 0 | 2 | 10 | | |
| Backwards Left | 7 | 2.7 | 1 | 2.2 | 1 | 2.2 | 2 | 3.4 | 1 | 2.3 | 1 | 3.6 | 1 | 5.3 | 0 | 0 | | |
| Right | 8 | 3.1 | 4 | 8.7 | 1 | 2.2 | 0 | 0 | 0 | 0 | 1 | 3.6 | 0 | 0 | 2 | 10 | | |
| Left | 14 | 5.4 | 4 | 8.7 | 3 | 6.5 | 1 | 1.7 | 2 | 4.5 | 1 | 3.6 | 1 | 5.3 | 2 | 10 | | |
| Tai Sabaki | 59 | 22.6 | 7 | 15.2 | 11 | 23.9 | 10 | 17.2 | 19 | 43.2 | 4 | 14.3 | 5 | 26.3 | 3 | 15 | | |
| Static | 123 | 47.1 | 19 | 41.3 | 23 | 50 | 32 | 55.2 | 18 | 40.9 | 18 | 64.3 | 6 | 31.6 | 7 | 35 | | |
| NSJ Movement | | | | | | | | | | | | | | | | | 75.939 | 0.026 |
| Forward | 16 | 6.1 | 6 | 13 | 2 | 4.3 | 4 | 6.9 | 1 | 2.3 | 1 | 3.6 | 1 | 5.3 | 1 | 5 | | |
| Forward Right | 7 | 2.7 | 1 | 2.2 | 1 | 2.2 | 2 | 3.4 | 1 | 2.3 | 1 | 3.6 | 1 | 5.3 | 0 | 0 | | |
| Forward Left | 7 | 2.7 | 0 | 0 | 2 | 4.3 | 3 | 5.2 | 0 | 0 | 0 | 0 | 0 | 0 | 2 | 10 | | |
| Backwards | 16 | 6.1 | 2 | 4.3 | 2 | 4.3 | 4 | 6.9 | 2 | 4.5 | 2 | 7.1 | 4 | 21.1 | 0 | 0 | | |
| Backwards Right | 7 | 2.7 | 1 | 2.2 | 1 | 2.2 | 1 | 1.7 | 1 | 2.3 | 0 | 0 | 0 | 0 | 3 | 15 | | |
| Backwards Left | 4 | 1.5 | 2 | 4.3 | 0 | 0 | 1 | 1.7 | 0 | 0 | 0 | 0 | 1 | 5.3 | 0 | 0 | | |
| Right | 14 | 5.4 | 4 | 8.7 | 3 | 6.5 | 1 | 1.7 | 2 | 4.5 | 1 | 3.6 | 1 | 5.3 | 2 | 10 | | |
| Left | 8 | 3.1 | 4 | 8.7 | 1 | 2.2 | 0 | 0 | 0 | 0 | 1 | 3.6 | 0 | 0 | 2 | 10 | | |
| Tai Sabaki | 59 | 22.6 | 7 | 15.2 | 11 | 23.9 | 10 | 17.2 | 19 | 43.2 | 4 | 14.3 | 5 | 26.3 | 3 | 15 | | |
| Static | 123 | 47.1 | 19 | 41.3 | 23 | 50 | 32 | 55.2 | 18 | 40.9 | 18 | 64.3 | 6 | 31.6 | 7 | 35 | | |
| Grouped techniques | | | | | | | | | | | | | | | | | 51.551 | 0.000 |
| Te waza | 82 | 31.4 | 18 | 39.1 | 14 | 30.4 | 25 | 43.1 | 6 | 13.6 | 7 | 25.0 | 5 | 26.3 | 7 | 35.0 | | |
| Koshi waza | 16 | 6.1 | 3 | 6.5 | 0 | 0.0 | 2 | 3.4 | 10 | 22.7 | 1 | 3.6 | 0 | 0.0 | 0 | 0.0 | | |
| Ashi waza | 86 | 33.0 | 16 | 34.8 | 13 | 28.3 | 11 | 19.0 | 15 | 34.1 | 15 | 53.6 | 6 | 31.6 | 10 | 50.0 | | |
| Sutemi waza | 77 | 29.5 | 9 | 19.6 | 19 | 41.3 | 20 | 34.5 | 13 | 29.5 | 5 | 17.9 | 8 | 42.1 | 3 | 15.0 | | |
| Individual techniques | | | | | | | | | | | | | | | | | 307.451 | 0.001 |
| De-ashi-harai | 11 | 4.2 | 3 | 6.5 | 0 | 0 | 1 | 1.7 | 3 | 6.8 | 2 | 7.1 | 1 | 5.3 | 1 | 5 | | |
| Harai-gosh | 2 | 0.8 | 0 | 0 | 0 | 0 | 0 | 0 | 2 | 4.5 | 0 | 0 | 0 | 0 | 0 | 0 | | |
| Harai-makikomi | 6 | 2.3 | 2 | 4.3 | 0 | 0 | 2 | 3.4 | 0 | 0 | 0 | 0 | 1 | 5.3 | 1 | 5 | | |
| Hikikomi-gaeshi | 9 | 3.4 | 3 | 6.5 | 2 | 4.3 | 3 | 5.2 | 0 | 0 | 0 | 0 | 0 | 0 | 1 | 5 | | |
| Hiza-guruma | 1 | 0.4 | 0 | 0 | 1 | 2.2 | 0 | 0 | 0 | 0 | 0 | 0 | 0 | 0 | 0 | 0 | | |
| Ippon-seoi-nage | 10 | 3.8 | 0 | 0 | 1 | 2.2 | 2 | 3.4 | 1 | 2.3 | 4 | 14.3 | 2 | 10.5 | 0 | 0 | | |
| Kata-guruma | 8 | 3.1 | 2 | 4.3 | 1 | 2.2 | 5 | 8.6 | 0 | 0 | 0 | 0 | 0 | 0 | 0 | 0 | | |
| Koshi-guruma | 4 | 1.5 | 0 | 0 | 0 | 0 | 0 | 0 | 4 | 9.1 | 0 | 0 | 0 | 0 | 0 | 0 | | |
| Ko-soto-gake | 11 | 4.2 | 0 | 0 | 4 | 8.7 | 1 | 1.7 | 3 | 6.8 | 1 | 3.6 | 2 | 10.5 | 0 | 0 | | |
| Ko-soto-gari | 3 | 1.1 | 0 | 0 | 2 | 4.3 | 1 | 1.7 | 0 | 0 | 0 | 0 | 0 | 0 | 0 | 0 | | |
| Ko-uchi-gari | 6 | 2.3 | 0 | 0 | 2 | 4.3 | 1 | 1.7 | 2 | 4.5 | 1 | 3.6 | 0 | 0 | 0 | 0 | | |
| Ko-uchi-makikomi | 1 | 0.4 | 0 | 0 | 1 | 2.2 | 0 | 0 | 0 | 0 | 0 | 0 | 0 | 0 | 0 | 0 | | |
| O-soto-gaeshi | 1 | 0.4 | 0 | 0 | 0 | 0 | 0 | 0 | 0 | 0 | 0 | 0 | 1 | 5.3 | 0 | 0 | | |
| O-soto-gari | 2 | 0.8 | 0 | 0 | 0 | 0 | 0 | 0 | 1 | 2.3 | 1 | 3.6 | 0 | 0 | 0 | 0 | | |
| O-soto-makikomi | 1 | 0.4 | 0 | 0 | 0 | 0 | 0 | 0 | 1 | 2.3 | 0 | 0 | 0 | 0 | 0 | 0 | | |
| O-soto-otoshi | 6 | 2.3 | 3 | 6.5 | 1 | 2.2 | 1 | 1.7 | 1 | 2.3 | 0 | 0 | 0 | 0 | 0 | 0 | | |
| O-uchi-gaeshi | 4 | 1.5 | 0 | 0 | 0 | 0 | 1 | 1.7 | 0 | 0 | 2 | 7.1 | 0 | 0 | 1 | 5 | | |
| O-uchi-gari | 11 | 4.2 | 4 | 8.7 | 0 | 0 | 1 | 1.7 | 2 | 4.5 | 1 | 3.6 | 1 | 5.3 | 2 | 10 | | |
| Sasae-tsurikomi-ashi | 9 | 3.4 | 0 | 0 | 2 | 4.3 | 2 | 3.4 | 0 | 0 | 1 | 3.6 | 0 | 0 | 4 | 20 | | |
| Seoi-nage | 20 | 7.7 | 6 | 13 | 2 | 4.3 | 11 | 19 | 1 | 2.3 | 0 | 0 | 0 | 0 | 0 | 0 | | |
| Seoi-otoshi | 7 | 2.7 | 2 | 4.3 | 2 | 4.3 | 2 | 3.4 | 1 | 2.3 | 0 | 0 | 0 | 0 | 0 | 0 | | |
| Sode-tsurikomi-goshi | 1 | 0.4 | 0 | 0 | 0 | 0 | 0 | 0 | 0 | 0 | 1 | 3.6 | 0 | 0 | 0 | 0 | | |
| Soto-makikomi | 3 | 1.1 | 0 | 0 | 0 | 0 | 1 | 1.7 | 1 | 2.3 | 0 | 0 | 1 | 5.3 | 0 | 0 | | |
| Sumi-gaeshi | 7 | 2.7 | 0 | 0 | 3 | 6.5 | 2 | 3.4 | 1 | 2.3 | 0 | 0 | 1 | 5.3 | 0 | 0 | | |
| Sumi-otoshi | 26 | 10 | 6 | 13 | 7 | 15.2 | 3 | 5.2 | 1 | 2.3 | 2 | 7.1 | 2 | 10.5 | 5 | 25 | | |
| Tai-otoshi | 9 | 3.4 | 2 | 4.3 | 1 | 2.2 | 0 | 0 | 2 | 4.5 | 1 | 3.6 | 1 | 5.3 | 2 | 10 | | |
| Tani-otoshi | 12 | 4.6 | 1 | 2.2 | 4 | 8.7 | 2 | 3.4 | 3 | 6.8 | 2 | 7.1 | 0 | 0 | 0 | 0 | | |
| Tomoe-nage | 15 | 5.7 | 2 | 4.3 | 3 | 6.5 | 6 | 10.3 | 1 | 2.3 | 0 | 0 | 3 | 15.8 | 0 | 0 | | |
| Tsubame-gaeshi | 1 | 0.4 | 0 | 0 | 1 | 2.2 | 0 | 0 | 0 | 0 | 0 | 0 | 0 | 0 | 0 | 0 | | |
| Tsuri-goshi | 4 | 1.5 | 1 | 2.2 | 0 | 0 | 1 | 1.7 | 2 | 4.5 | 0 | 0 | 0 | 0 | 0 | 0 | | |
| Tsurikomi-goshi | 2 | 0.8 | 1 | 2.2 | 0 | 0 | 0 | 0 | 1 | 2.3 | 0 | 0 | 0 | 0 | 0 | 0 | | |
| Uchi-makikomi | 1 | 0.4 | 0 | 0 | 0 | 0 | 0 | 0 | 0 | 0 | 0 | 0 | 0 | 0 | 1 | 5 | | |
| Uchi-mata | 22 | 8.4 | 6 | 13 | 0 | 0 | 4 | 6.9 | 4 | 9.1 | 6 | 21.4 | 1 | 5.3 | 1 | 5 | | |
| Uchi-mata-makikomi | 2 | 0.8 | 1 | 2.2 | 0 | 0 | 0 | 0 | 1 | 2.3 | 0 | 0 | 0 | 0 | 0 | 0 | | |
| Uki-goshi | 1 | 0.4 | 0 | 0 | 0 | 0 | 0 | 0 | 0 | 0 | 0 | 0 | 0 | 0 | 1 | 5 | | |
| Ura-nage | 8 | 3.1 | 0 | 0 | 3 | 6.5 | 1 | 1.7 | 1 | 2.3 | 2 | 7.1 | 1 | 5.3 | 0 | 0 | | |
| Ushiro-goshi | 1 | 0.4 | 0 | 0 | 0 | 0 | 1 | 1.7 | 0 | 0 | 0 | 0 | 0 | 0 | 0 | 0 | | |
| Utsuri-goshi | 2 | 0.8 | 1 | 2.2 | 0 | 0 | 0 | 0 | 1 | 2.3 | 0 | 0 | 0 | 0 | 0 | 0 | | |
| Yoko-gake | 4 | 1.5 | 0 | 0 | 2 | 4.3 | 0 | 0 | 1 | 2.3 | 0 | 0 | 1 | 5.3 | 0 | 0 | | |
| Yoko-otoshi | 7 | 2.7 | 0 | 0 | 1 | 2.2 | 3 | 5.2 | 2 | 4.5 | 1 | 3.6 | 0 | 0 | 0 | 0 | | |
| Score | | | | | | | | | | | | | | | | | 13.760 | 0.032 |
| Ippon | 95 | 36.4 | 16 | 34.8 | 14 | 30.4 | 13 | 22.4 | 17 | 38.6 | 16 | 57.1 | 10 | 52.6 | 9 | 45 | | |
| Wazari | 166 | 63.6 | 30 | 65.2 | 32 | 69.6 | 45 | 77.6 | 27 | 61.4 | 12 | 42.9 | 9 | 47.4 | 11 | 55 | | |

Note: Fr. = Frequency; SJ = Scoring judoka; NSJ = Non-scoring judoka.

The distribution of scoring actions across different minutes of combat was relatively homogeneous among the various weight categories, with no significant differences observed ($\chi^2 = 19.468$, $p = 0.727$). It is noted that the majority of scoring actions occurred in the initial minutes of combat across all weight categories. Conversely, few scoring actions were recorded during the final minutes of the combat. In four weight categories (−66 kg, −90 kg, −100 kg, and +100 kg), no scoring actions were observed during the Golden Score. Moreover, no scoring actions were recorded in the last minute of combat in the +100 kg category.

The distribution of scoring actions based on the partial result of the combat is quite homogeneous across the different weight categories. No significant differences were found ($\chi^2 = 14.295$, $p = 0.282$). Most scoring actions occur when the score is tied, exceeding 65% in all weight categories. There are few scoring actions recorded when a judoka is trailing.

The distribution of scoring actions based on the penalties of the scoring judoka is quite homogeneous across different weight categories, as no significant differences were found among them ($\chi^2 = 11.196$, $p = 0.512$). It is observed that the majority of scoring actions occur when the judoka has no penalties, exceeding 60% across all weight categories. Scoring actions are rare when the scoring judoka has two shidos, representing only 8.4% of the total scoring actions.

The distribution of scoring actions based on the penalties of the judoka who receives the point is fairly homogeneous across the different weight categories, with no significant differences ($\chi^2 = 9.59$, $p = 0.652$). It is observed that most scoring actions occur when the judoka has no shido. There are hardly any scoring actions recorded when the judoka who does not score has two shidos.

The distribution of scoring actions based on the grip of the scoring judoka is homogeneous across the different weight categories, with no significant differences ($\chi^2 = 74.541$, $p = 0.098$). Most scoring actions occur when the judoka uses a lapel-sleeve grip, accounting for 60.9% of the actions. The sleeve-back grip is the second most common, with 15.3%. Other grips do not exceed 7% individually.

The distribution of scoring actions according to the grip of the non-scoring judoka is homogeneous across the different weight categories, as no significant differences were found ($\chi^2 = 60.953$, $p = 0.441$). The majority of scoring actions occur when the judoka uses a lapel-sleeve grip, accounting for 78.9% of the actions. The sleeve-back grip is the second most common, at 6.9%. The other grips do not exceed 5% individually.

The distribution of scoring actions based on the movement of the scoring judoka has shown to be heterogeneous across different weight categories, with significant differences observed ($\chi^2 = 75.939$, $p = 0.026$). The effect size test indicated that the relationship was moderate ($V = 0.220$). In general, the majority of scoring actions occur in a static position, accounting for 47.1% of the total. Tai sabaki is the next most common movement, representing 22.6%. Notably, in the −81 kg category, actions in tai sabaki exceed those in a static position (corrected residual 3.6), while in the −100 kg category, they are very similar.

The distribution of scoring actions based on the movement of the non-scoring judoka is heterogeneous among the different weight categories, with significant differences observed ($\chi^2 = 75.939$, $p = 0.026$). The effect size test indicated that the relationship was moderate ($V = 0.220$). Generally, most scoring actions occur when the judoka is stationary (47.1%), followed by actions in tai sabaki (22.6%). Notably, in the −81 kg category, the actions in tai sabaki exceed those in a static position (corrected residual 3.6), while in the −100 kg category, they are very similar.

The distribution of scoring actions based on the technique group used by the scoring judoka is heterogeneous among the different weight categories. Significant differences were found ($\chi^2 = 51.551$, $p = 0.0004$). The effect size test indicated a moderate relationship ($V = 0.257$). In general, there is a similar average usage among the groups of te waza, ashi waza, and sutemi waza, with koshi waza being the least utilized. By weight categories, clear differences are observed. In the 73 kg category, arm techniques are the most frequently used (adjusted residual 2.2), while leg techniques are the least utilized (adjusted residual −2.6). In the 81 kg category, hip techniques are more common (adjusted residual 5), while arm

techniques are less utilized (adjusted residual −2.8). In the 90 kg category, leg techniques are the most frequently used (adjusted residual 2.5).

The distribution of scoring actions based on the technique used by the scoring judoka was heterogeneous across different weight categories. Significant differences were found between them ($\chi^2 = 307.451$, $p = 0.001$). The effect size test indicated a strong relationship ($V = 0.443$). Each technique is used differently in each weight category. In the 60 kg category, the most used techniques are seoi-nage, sumi-otoshi, and uchi-mata (all three with equal frequency), and o-soto-otoshi has a higher than expected frequency (adjusted residual 2.1). In the 66 kg category, the most frequently used technique is sumi-otoshi. Techniques like hiza-guruma, ko-soto-gari, ko-uchi-makikomi, and tsubame-gaeshi have a higher than expected frequency (all with adjusted residuals of 2.2), while uchi-mata has a lower than expected frequency (adjusted residual −2.3). In the 73 kg category, the most used technique is seoi-nage, which also has a higher than expected frequency (adjusted residual 3.7), as does kata-guruma (adjusted residual 2.8). In the 81 kg category, the most used techniques are koshi-guruma and uchi-mata (both equally frequent). Three techniques have a higher than expected frequency: harai-goshi, koshi-guruma, and o-soto-makikomi, with adjusted residuals of 3.2, 4.5, and 2.2, respectively. In the 90 kg category, the most used technique is uchi-mata. Four techniques have a higher than expected frequency: ippon-seoi-nage, o-uchi-gaeshi, sode-tsurikomi-goshi, and uchi-mata, with adjusted residuals of 3.1, 2.6, 2.9, and 2.6, respectively. In the 100 kg category, the most used technique is tomoe-nage, which, along with o-soto-gaeshi, has a higher than expected frequency (adjusted residuals of 3.6 and 2, respectively). Finally, in the +100 kg category, the most used technique is sumi-otoshi, and four techniques have a higher than expected frequency: sasae-tsurikomi-ashi, sumi-otoshi, uchi-makikomi, and uki-goshi, with adjusted residuals of 4.2, 2.3, 3.5, and 3.5, respectively.

The distribution of scoring actions based on the score achieved was heterogeneous across different weight categories, with significant differences between them ($\chi^2 = 13.76$, $p = 0.032$). The effect size test indicated a moderate relationship ($V = 0.230$). A greater number of waza-aris was observed in lower weight categories compared to higher weight categories, where ippons matched or even exceeded the number of waza-aris. Specifically, in the 73 kg category, waza-ari had a higher-than-expected frequency (corrected residual 2.5), while in the 90 kg category, ippon had a higher-than-expected frequency (corrected residual 2.4).

Table S1 presents the descriptive analysis of scoring actions in groundwork judo (n = 52). For further details, see Table S1 in the Supplementary Materials.

The distribution of scoring actions based on the time of the combat was heterogeneous across the different weight categories, with significant differences between them ($\chi^2 = 36.683$, $p = 0.047$). The effect size test indicated a strong relationship ($V = 0.420$). Generally, there were more scoring actions during the initial minutes of the combat in most weight categories. In heavier weight categories, there were almost no scoring actions in ground judo during the final minutes of the combat. In the −60 kg category, more scoring actions than expected occurred in the third minute (adjusted residual 4.1). In the −66 kg and −73 kg categories, more scoring actions than expected were observed in the fourth minute of the combat (both with an adjusted residual of 2).

The distribution of scoring actions based on the partial outcome of the combat was homogeneous across different weight categories, as no significant differences were observed between them ($\chi^2 = 10.611$, $p = 0.563$). A higher number of scoring actions occurred when the score was tied or when the competitor was winning. Scoring actions were rare when the competitor was losing the combat.

The distribution of scoring actions based on the penalties of the scoring judoka was homogeneous across different weight categories, as no significant differences were found between them ($\chi^2 = 13.6$, $p = 0.327$). A higher number of scoring actions occurred when the scoring judoka had no penalties, with some weight categories (−90 and −100 kg)

only showing scoring actions under these conditions. Scoring actions were rare when the competitor had one shido and very rare with two shidos.

The distribution of scoring actions based on the penalties of the judoka who does not score was homogeneous across different weight categories, as no significant differences were found between them ($\chi^2 = 9.974$, $p = 0.618$). A greater number of scoring actions occurred when the non-scoring judoka had no penalties. Scoring actions were rare when the non-scoring judoka had one shido and very rare with two shidos.

The distribution of scoring actions based on the position of the judoka who scores on the ground is quite clear, with little variation among weight categories. No significant differences were found ($\chi^2 = 56.288$, $p = 0.389$). Generally, most scoring actions occur when the judoka who scores is in a favorable position or mounted on top.

The distribution of scoring actions based on the situation of the non-scoring judoka on the ground is homogeneous across different weight categories, as no significant differences were found ($\chi^2 = 48.663$, $p = 0.446$). A higher number of scoring actions occurs when the non-scoring judoka is in an unfavorable position, in all fours, or in a prone position.

The distribution of scoreable actions based on the action of the judoka who scores on the ground is heterogeneous among the different weight categories, as significant differences were found ($\chi^2 = 36.744$, $p = 0.046$). The effect size test indicated that the relationship was strong ($V = 0.420$). Generally, a higher number of scoreable actions occur when the judoka who scores performs a combination, a direct action, or a turnover. In the −60 kg category, the most frequently used action is the turnover, which also has a frequency greater than expected (corrected residual 2.3). Conversely, the direct action has a frequency lower than expected (corrected residual −2.1). In the −66 kg, −73 kg, −90 kg, and −100 kg categories, the most frequently used action is direct action. In the −90 kg category, both the direct action and the action of withdrawing the entangled leg with the free leg have a frequency greater than expected (corrected residuals 2.2 and 2, respectively). Finally, in the −81 kg and +100 kg categories, the most frequently used action is the combination. In the −81 kg category, this action has a frequency greater than expected (corrected residual 2.2), while the direct action has a frequency lower than expected (corrected residual −2.1).

The distribution of scoring actions based on the technique group employed by the judoka who scores on the ground is heterogeneous across the different weight categories. Significant differences were found among these categories ($\chi^2 = 23.37$, $p = 0.025$). The effect size test indicated that the relationship was strong ($V = 0.474$). In general, a greater number of scoring actions occur when the scoring judoka employs a technique from the osaekomi waza group. In some weight categories, this is the only technique used to score (−60 kg, −81 kg, and +100 kg). Specifically, these techniques had a higher than expected frequency in the +100 kg category (corrected residual 2). Strangulations also proved effective in the −66 kg, −90 kg, and −100 kg categories, with a higher than expected frequency in the −66 kg category (corrected residual 2.2). Finally, joint locks were only useful in the −73 kg and −90 kg categories, exhibiting a higher than expected frequency in the −73 kg category (corrected residual 2.5).

The distribution of scoring actions based on the technique used by the judoka who scores on the ground is homogeneous across different weight categories. No significant differences were found among these ($\chi^2 = 85.18$, $p = 0.056$). A greater number of scoring actions occurs when the scoring judoka employs a technique of yoko-shiho-gatame or tate-shiho-gatame from the osaekomi waza group, or okuri-eri-jime from the shime-waza group.

The distribution of scoring actions based on the score following a scoring action on the ground is homogeneous across the different weight categories, as no significant differences were found among them ($\chi^2 = 10.401$, $p = 0.109$). In general, there are more scoring actions that result in ippon than in wazari in all weight categories, except for 81 kg and +100 kg, where more wazari are achieved, especially in the 81 kg category.

### 3.3. Analysis Using T-Patterns

The search for T-patterns was conducted based on the following criteria: First, a search was carried out from an individual perspective. This perspective corresponds to the different categories of the observational instrument. In particular, the criterion "technique" includes all the techniques classified by Kodokan (68 throws and 32 control techniques). This implies a high dispersion of data, which considerably reduces the possibility of detecting patterns. For this reason, secondly, a search for patterns was conducted from a grouped perspective. The 100 techniques were grouped according to the categories established by Kodokan: arm techniques (te waza), leg techniques (ashi), hip techniques (koshi), sacrifice techniques (sutemi), hold-down techniques (osaekomi), strangulations (shime), and joint locks (kansetsu). With this strategy, the dispersion of the data was reduced from 100 possible patterns to 7, favoring the detection of patterns. In both cases, a search for patterns with a minimum of three occurrences in all weight categories was conducted. This search and grouping strategy has been employed in previous studies [29]. The results of these searches are presented in Table 4.

**Table 4.** Quantity and type of patterns according to different search strategies.

| | Individual Search | | | | | | | Grouped Techniques Search | | | | | | |
|---|---|---|---|---|---|---|---|---|---|---|---|---|---|---|
| | −60 kg | −66 kg | −73 kg | −81 kg | −90 kg | −100 kg | +100 kg | −60 kg | −66 kg | −73 kg | −81 kg | −90 kg | −100 kg | +100 kg |
| Total patterns (n) | 170 | 181 | 216 | 172 | 104 | 44 | 52 | 220 | 215 | 287 | 189 | 121 | 47 | 55 |
| Discarded patterns (n) | 165 | 174 | 210 | 171 | 104 | 43 | 48 | 169 | 186 | 220 | 170 | 115 | 45 | 49 |
| SA Patterns (n) | 5 | 7 | 6 | 1 | 0 | 1 | 4 | 51 | 29 | 67 | 19 | 6 | 2 | 6 |
| Ippon | 1 | 2 | 0 | 0 | 0 | 1 | 0 | 17 | 10 | 4 | 5 | 6 | 2 | 3 |
| Waza Ari | 4 | 5 | 6 | 1 | 0 | 0 | 4 | 34 | 19 | 63 | 14 | 0 | 0 | 3 |
| Ashi | 4 | 1 | | | | | | 29 | 12 | 20 | 15 | 6 | | 2 |
| Te | 1 | 4 | 5 | | | | 1 | 17 | 7 | 27 | | | | 1 |
| Koshi | | | | | | | | | | | 2 | | | |
| Sutemi | | | 1 | | | | | 2 | 7 | 20 | 1 | | | |
| Osaekomi | | | | 1 | | | 3 | 3 | 1 | | 1 | | 1 | 3 |
| Kansetsu | | | | | | | | | | | | | | |
| Shime | | 2 | | | | 1 | | | 2 | | | | 1 | |
| Direct attack standing | 4 | 4 | 6 | | | | 1 | 46 | 21 | 67 | 18 | 3 | | 3 |
| Direct attack ground | | 2 | | | | 1 | 2 | 3 | 3 | | | | 1 | 2 |
| Combination | | | | | | | | 1 | | | | 2 | | |
| Counterattack | 1 | 1 | | | | | | 1 | 5 | | | 1 | 1 | |
| Transition technique | | | | 1 | | | 1 | | | | 1 | | | 1 |

Note: SA = Score Action.

From an individual perspective, several relevant observations can be made. Firstly, the number of detected scoring action patterns is limited. In the −90 kg category, no scoring action patterns were identified. In the −60, −66, −73, −81, and +100 kg categories, waza-ari patterns are more numerous than ippon patterns; however, this is not the case in the −90 and −100 kg categories. From a grouped perspective of the techniques, this trend is further emphasized. It is observed that the heavier weight categories (−90, −100, and +100 kg) follow a different trend compared to the other categories.

Regarding the type of technique, it is observed that the patterns of standing techniques are significantly more numerous than the patterns of ground techniques. In standing techniques, the most commonly used patterns involve leg and arm techniques. Patterns of hip techniques are only present in the −81 kg category (grouped perspective). Patterns of sacrifice techniques are particularly relevant in the −73 kg category (grouped perspective). This distribution may vary depending on the weight category; in the −81, −90, and −100 kg categories, patterns of scoring actions with arm techniques are absent. Additionally, it is noteworthy that in ground fighting, there are no patterns of scoring actions with joint lock techniques, with immobilizations being the most utilized.

Regarding the type of action, it is observed that, in general, scoring is mainly achieved through a pattern of direct attack in standing position. This pattern is very frequent in the lighter weight categories and becomes less common in the heavier categories. Other actions, such as direct attacks on the ground, combinations, counterattacks, and linkages, are patterns that are infrequently utilized.

Table S2 presents the selected patterns from an individual perspective of the technique. These patterns are related to our research focus. For further details, see Table S2 in the Supplementary Materials.

Table 5 presents the selected patterns from a grouped perspective of the technique.

**Table 5.** T-patterns of scoring actions with grouped techniques.

| DIRECT ATTACKS | | |
|---|---|---|
| **Standing direct attacks** | **O** | **I** |
| 60 Kg | | |
| Te-waza | | |
| (even ((standing nsj,lapel-sleeve)(sj,lapel-sleeve sj,te,ippon))) | 4 | 1 |
| (3rd-minute ((standing nsj,lapel-sleeve)(sj,lapel-sleeve sj,te,ippon))) | 3 | 2 |
| ((not-shido (standing nsj,lapel-sleeve))(sj,lapel-sleeve sj,te,ippon)) | 3 | 3 |
| ((standing nsj,lapel-sleeve)(sj,lapel-sleeve sj,te,ippon)) | 6 | 4 |
| (standing ((nsj,lapel-sleeve sj,static) sj,te,ippon)) | 3 | 5 |
| (((not-shido (standing nsj,lapel-sleeve)) sj,lapel-sleeve) sj,te,waza-ari) | 4 | 6 |
| ((standing nsj,lapel-sleeve)(sj,tai-sabaki sj,te,waza-ari)) | 3 | 7 |
| Ashi-waza | | |
| ((2nd-minute ((even standing)(nsj,lapel-sleeve sj,lapel-sleeve))) sj,ashi,ippon) | 4 | 8 |
| ((even standing)(nsj,lapel-sleeve ((sj,lapel-sleeve sj,static) sj,ashi,ippon))) | 3 | 9 |
| (shido2,sj ((standing nsj,lapel-sleeve)(sj,lapel-sleeve sj,ashi,waza-ari))) | 3 | 10 |
| (shido2,nsj ((standing nsj,lapel-sleeve)(sj,lapel-sleeve sj,ashi,waza-ari))) | 3 | 11 |
| ((even standing)((nsj,lapel-sleeve sj,lapel-sleeve)(sj,static sj,ashi,waza-ari))) | 3 | 12 |
| (sj,static sj,ashi,waza-ari) | 5 | 13 |
| Sutemi-waza | | |
| ((standing nsj,lapel-sleeve) sj,sutemi,ippon) | 3 | 14 |
| (sj,tai-sabaki sj,sutemi,waza-ari) | 3 | 15 |
| 66 Kg | | |
| Te-waza | | |
| ((win (standing nsj,lapel-sleeve)) sj,te,waza-ari) | 4 | 16 |
| (((3rd-minute standing) nsj,lapel-sleeve) sj,te,waza-ari) | 3 | 17 |
| (nsj,static sj,te,waza-ari) | 4 | 18 |
| Ashi-waza | | |
| ((1st-minute even)((not-shido (standing nsj,lapel-sleeve))(sj,lapel-sleeve sj,ashi,ippon))) | 3 | 19 |
| ((nsj,lapel-sleeve sj,lapel-sleeve)(sj,static sj,ashi,ippon)) | 3 | 20 |
| (sj,static sj,ashi,ippon) | 4 | 21 |
| (standing (sj,static sj,ashi,waza-ari)) | 4 | 22 |
| Sutemi-waza | | |
| ((even not-shido)(standing ((nsj,lapel-sleeve sj,lapel-sleeve)(sj,static sj,sutemi,waza-ari)))) | 3 | 23 |
| ((2nd-minute (not-shido standing)) sj,sutemi,waza-ari) | 6 | 24 |
| 73 Kg | | |
| Te-waza | | |
| ((not-shido (standing (nsj,lapel-sleeve sj,lapel-sleeve))) sj,te,ippon) | 4 | 25 |
| ((even (standing nsj,lapel-sleeve))(sj,lapel-sleeve (sj,static sj,te,waza-ari))) | 3 | 26 |
| ((2nd-minute not-shido)((standing sj,static) sj,te,waza-ari)) | 3 | 27 |
| (2nd-minute ((standing sj,static) sj,te,waza-ari)) | 4 | 28 |
| ((not-shido (standing sj,static)) sj,te,waza-ari) | 5 | 29 |
| ((standing sj,static) sj,te,waza-ari) | 7 | 30 |
| ((standing sj,tai-sabaki) sj,te,waza-ari) | 3 | 31 |
| Ashi-waza | | |
| (standing (nsj,lapel-sleeve (sj,lapel-sleeve sj,ashi,ippon))) | 3 | 32 |
| ((standing (nsj,lapel-sleeve sj,lapel-sleeve))(sj,static sj,ashi,waza-ari)) | 5 | 33 |
| ((shido1,sj standing)(nsj,lapel-sleeve sj,ashi,waza-ari)) | 4 | 34 |
| ((shido1,nsj standing)(nsj,lapel-sleeve sj,ashi,waza-ari)) | 4 | 35 |
| Sutemi-waza | | |
| (standing (nsj,lapel-sleeve (sj,lapel-sleeve sj,sutemi,ippon))) | 3 | 36 |
| ((standing (nsj,lapel-sleeve sj,lapel-sleeve)) sj,sutemi,waza-ari) | 5 | 37 |
| (((1st-minute (not-shido standing)) sj,lapel-back) sj,sutemi,waza-ari) | 4 | 38 |
| (standing (sj,tai-sabaki sj,sutemi,waza-ari)) | 5 | 39 |
| (sj,static sj,sutemi,waza-ari) | 4 | 40 |
| 81 Kg | | |
| Ashi-waza | | |
| (((2nd-minute win)(standing (nsj,lapel-sleeve sj,lapel-sleeve))) sj,ashi,ippon) | 3 | 46 |
| ((win standing)((nsj,lapel-sleeve sj,lapel-sleeve) sj,ashi,ippon)) | 4 | 47 |
| (standing ((nsj,lapel-sleeve sj,lapel-sleeve) sj,ashi,ippon)) | 5 | 48 |
| ((even (standing nsj,lapel-sleeve))(sj,lapel-sleeve (sj,tai-sabaki sj,ashi,waza-ari))) | 3 | 49 |
| ((even (standing nsj,lapel-sleeve))(sj,lapel-sleeve sj,ashi,waza-ari)) | 5 | 50 |
| (1st-minute ((even (standing sj,static)) sj,ashi,waza-ari)) | 3 | 51 |
| (standing sj,ashi,waza-ari) | 8 | 52 |

**Table 5.** *Cont.*

| DIRECT ATTACKS | | |
|---|---|---|
| **Standing direct attacks** | **O** | **I** |
| Koshi-waza | | |
| ((win (standing nsj,lapel-sleeve)) sj,koshi,ippon) | 3 | 53 |
| ((even (standing nsj,lapel-sleeve))(sj,tai-sabaki sj,koshi,waza-ari)) | 3 | 54 |
| Sutemi-waza | | |
| (((not-shido (standing nsj,lapel-sleeve)) sj,lapel-sleeve) sj,sutemi,ippon) | 3 | 55 |
| 90 Kg | | |
| Ashi-waza | | |
| (even (((standing nsj,lapel-sleeve)(sj,lapel-sleeve sj,static)) sj,ashi,ippon)) | 4 | 41 |
| (((standing nsj,lapel-sleeve)(sj,lapel-sleeve sj,static)) sj,ashi,ippon) | 5 | 42 |
| M100 Kg | | |
| Ashi-waza | | |
| (((1st-minute even)(not-shido ((standing nsj,lapel-sleeve) sj,static))) sj,ashi,ippon) | 3 | 43 |
| (((1st-minute even)(not-shido (standing sj,lapel-sleeve))) sj,ashi,ippon) | 4 | 44 |
| Te-waza | | |
| (((2nd-minute even)(standing nsj,lapel-sleeve)) sj,te,waza-ari) | 3 | 45 |
| **Ground direct attacks** | **O** | **I** |
| 60 Kg | | |
| Osaekomi-waza | | |
| ((even (ground nsj,all-fours-position))(sj,rolling-action sj,osaekomi,ippon)) | 3 | 56 |
| 66 Kg | | |
| Osaekomi-waza | | |
| (ground sj,osaekomi,ippon) | 3 | 57 |
| Shime-waza | | |
| ((ground sj,lateral-position)(sj,direct-action sj,shime,ippon)) | 3 | 58 |
| (ground (sj,direct-action sj,shime,ippon)) | 4 | 59 |
| 100 Kg | | |
| Shime-waza | | |
| ((ground sj,mounted-same-direction)(sj,direct-action sj,shime,ippon)) | 3 | 60 |
| M100 Kg | | |
| Osaekomi-waza | | |
| (ground sj,osaekomi,ippon) | 4 | 61 |
| (ground sj,osaekomi,waza-ari) | 5 | 62 |
| **TRANSITION TECHNIQUE (Standing-Ground)** | **O** | **I** |
| 81 Kg | | |
| (ground ((sj,transition-technique nsj,inferior-position)(sj,frolling-action sj,osaekomi,waza-ari))) | 5 | 63 |
| M100 Kg | | |
| (ground ((sj,transition-technique nsj,inferior-position)(sj,frolling-action sj,osaekomi,waza-ari))) | 4 | 64 |
| **COMBINATION (Standing-Standing)** | **O** | **I** |
| 60 Kg | | |
| (shido2,nsj ((standing sj,lapel-sleeve)(sj,ashi sj,ashi,waza-ari))) | 3 | 65 |
| 90 Kg | | |
| (standing (nsj,lapel-sleeve ((sj,lapel-sleeve sj,static)(sj,ashi sj,ashi,ippon)))) | 3 | 66 |
| (sj,ashi sj,ashi,ippon) | 4 | 67 |
| **COUNTERATTACK** | **O** | **I** |
| 60 Kg | | |
| (nsj,ashi sj,te,waza-ari) | 3 | 68 |
| 66 Kg | | |
| ((standing nsj,lapel-sleeve)((sj,lapel-sleeve nsj,ashi) sj,te,waza-ari)) | 3 | 69 |
| (nsj,static (nsj,ashi sj,te,waza-ari)) | 3 | 70 |
| (nsj,ashi sj,te,waza-ari) | 5 | 71 |
| (nsj,static (nsj,ashi sj,sutemi,waza-ari)) | 3 | 72 |
| (nsj,ashi sj,sutemi,waza-ari) | 4 | 73 |
| 90 Kg | | |
| (nsj,ashi sj,ashi,ippon) | 3 | 74 |
| 100 Kg | | |
| (nsj,sutemi (ground sj,osaekomi,ippon)) | 4 | 75 |

Note: O = Occurrences; I = Identifier in the text; sj = scoring judoka; nsj = non-scoring judoka.

In the Supplementary Material, we describe the most relevant patterns from Tables S2 and 5.

## 4. Discussion

The aim of this study was to analyze the technical–tactical patterns of scoring actions in para-judo, as well as differences in performance based on weight and combat context. The results provide a clear insight into the trends and strategies used by judokas across different weight categories.

In Tachi-waza, it was observed that most points (75.5% of the total) are obtained in the first two minutes of the combat. This trend shows a progressive decrease in the number of scores as the combat progresses. Only five combats were registered during Golden Score. These findings are consistent with previous studies on the same championship, which analyze the duration of combats [17]. Moreover, it is notable that combats in higher weight categories tend to be shorter.

Regarding the partial score, it was found that judokas score more frequently when the score is tied. This makes sense, as the combat starts with an equal score. It was found that the probability of scoring is higher when leading than when trailing. This result is consistent with previous research on sighted judokas [29].

Regarding penalties, it was observed that judokas who score have similar percentages to those who do not score when performing actions that lead to scoring. This finding contrasts with previous studies indicating that judokas who receive scores tend to accumulate more shidos than those who produce them in judo for sighted athletes [34]. This suggests that, in judo for visually impaired athletes, shido might not be as significant as in judo for sighted athletes. This observation is supported by other authors [18].

The grips used by both judokas to score are predominantly sleeve-lapel grips. This makes sense, as it is the most common grip in judo for sighted athletes [35]. Additionally, this type of grip is used at the start and restart of the combat in Paralympic judo, which increases the likelihood that judokas will maintain it before risking losing the grip. The sleeve-back grip follows, which is a common progression from the sleeve-lapel grip.

In relation to movement, the principle of Ju, one of the fundamental principles of judo, states that one must use the opponent's force to throw them. This implies performing techniques while moving [36]. However, in our study, almost half of the scoring techniques were performed statically (47.1%), followed by movements in tai-sabaki. This finding is similar to what occurs in sighted judokas, where 48.2% of the scoring techniques were performed statically [29]. Another fundamental principle, seiryoku zenyo, suggests that if the opponent's strength cannot be used, one must take the initiative with the least possible effort [36]. This could explain the high percentage of techniques performed statically.

Regarding the distribution of techniques, it was observed that the Te-waza, Ashi-waza, and Sutemi-waza groups are relatively evenly distributed, each group accounting for about 30% of the actions. In contrast, Koshi-waza techniques are significantly less frequent, representing only 6% of the total. This distribution follows the general trend observed in judo for sighted athletes, although there are differences in percentages [8].

When analyzing each weight category, it was found that Te-waza techniques are more common in the lighter categories, while Ashi-waza techniques predominate in the heavier categories. This finding is consistent with previous studies on judo for sighted athletes [37]. However, in the −100 kg category, a notable trend toward the use of Sutemi-waza was observed, which could reflect specific characteristics of that category.

The diversity of techniques is considerable. This is due to the wide range of available techniques and the different ways of practicing judo, which can increase the chances of victory [21]. However, Seoi-nage, which is commonly used by sighted judokas [38], is identified as a highly efficient and frequently used technique, along with Uchi-mata and Sumi-otoshi. These are the only techniques performed more than 20 times.

The distribution of scores shows differences compared to other studies on judo for visually impaired athletes. In those studies, ippon represented 54.64% of the scores [18]. In our research, the percentage was only 36.4%.

When considering each weight category, the distribution of scores is similar to that observed in the Beijing 2008 Olympic Games. In the higher weight categories, there was a higher number of ippon, while in the lighter categories, a lower percentage was observed. This similarity is surprising given the change in regulations between both championships and the transition from judo for visually impaired athletes to judo for sighted athletes [38,39].

Regarding groundwork, the number of scoring actions decreases significantly compared to Tachi-waza, dropping from 261 to only 50 (16%). This decrease is considerable compared to judo for sighted athletes, where groundwork represents 25% of the scores [8]. As in Tachi-waza, most of the scoring actions in Ne-waza occur during the first two minutes of the combat, with only two cases registered in Golden Score. In 40% of the cases, the judoka who scores is winning at the time of the action. This may be because groundwork actions often follow a previous standing score, with 34.6% of the actions occurring directly after a previous standing score.

Penalties in Ne-waza follow a similar distribution to those observed in Tachi-waza for both the judokas who score and those who do not. This reinforces the idea that judo for visually impaired athletes is less affected by penalties. This phenomenon is corroborated by other authors [18], who found that penalties have a greater impact on Olympic judo than on Paralympic judo.

The predominant group of techniques in Ne-waza is Osaekomi-waza. This may be due to the high frequency with which judokas perform transition actions between standing and groundwork. The distribution of techniques shows a similar trend to that observed in judo for sighted athletes, but with a higher frequency of immobilizations in our research (73.1%) compared to previous studies, which showed 64% in 2015 and 76% in 2008. Additionally, the Shime-waza group represents 21.2% in our research, compared to 17% in 2008 and 4.25% in 2015. Conversely, the Kansetsu-waza group showed only 5.8% in our research, compared to 19% in 2008 and 2015 [8,38].

Unlike standing judo, in Ne-waza, the majority score is ippon, a trend also observed in sighted judokas [29]. This could be explained by the fact that, in 27% of the scoring techniques, the only possible score is ippon.

Regarding technical–tactical sequences, it was observed that there is not much transfer of movement patterns from one weight category to another. Each category presents distinct movement patterns. When observing these patterns without grouping techniques, very few movement patterns are identified. This reinforces the idea that a more varied judo produces better results [21].

When techniques are grouped, it is observed that movement patterns are more commonly found in the lighter weight categories. This could be due to the higher number of scores and participants compared to the heavier categories.

### 4.1. Practical Applications

The results of this study allow us to propose practical applications for the training and planning of judokas with visual impairments.

Most scoring actions occur in the first two minutes. Therefore, coaches and judokas should focus on high-intensity strategies from the start of the match. This maximizes scoring opportunities.

The results indicate that many throws are performed from a static position. It is essential for coaches to incorporate specific exercises to enhance effectiveness in this position. This optimizes the ability to attack without prior movement.

Transitions between standing techniques and hold-down techniques proved to be highly effective. Training fluidity in these transitions can increase scoring opportunities and promote more effective technical execution during the match.

Counterattacks and techniques showed variations in effectiveness depending on weight category. Personalizing training to adjust these patterns according to the judoka's weight will allow competitors to leverage techniques and combinations with a higher likelihood of success in their specific category.

### 4.2. Limitations of the Research and Future Perspectives

Until 2022, judokas with visual impairment were classified into three levels based on their degree of visual impairment (B1: blind; B2: severely impaired vision; and B3: moderate to poor vision). Recently, the IBSA modified the visual classification rules for

judo as well as the competition rules [40]. As a result, the visual categories (B1, B2, and B3) were reduced from three to two (J1 and J2): J1 for blind judokas and J2 for partially sighted judokas. In addition to modifying the visual categories, the competition format also changed. Now, judokas fight against opponents from the same visual category (J1 vs. J1 and J2 vs. J2). Previously, all athletes competed together, regardless of their visual classification. These modifications also led to an adjustment in weight categories [40]. Men's weight categories were reduced from seven (−60 kg, −66 kg, −73 kg, −81 kg, −90 kg, −100 kg, and +100 kg) to four (−60 kg, −73 kg, −90 kg, and +90 kg).

Due to these changes, we consider that the reduction in weight categories may have altered the pattern of scoring actions. Therefore, the results obtained in this study should be interpreted with caution. Additionally, this situation opens up new opportunities for future research.

Another future line of research would be to replicate this study in women, considering the new weight categories. The women's weight categories were reduced from six (−48 kg, −52 kg, −57 kg, −63 kg, −70 kg, and +70 kg) to four (−48 kg, −57 kg, −70 kg, and +70 kg).

Regarding judo for visually impaired athletes, it is crucial to highlight the impact of visual category on competitive performance. It has been found that para-judokas in category B1 (with total blindness) tend to perform worse compared to para-judokas in categories B2 or B3, who have partial residual vision [13,14]. These differences suggest that visual ability, even if limited, can be an important factor influencing the effectiveness of technical actions and scoring ability. Therefore, given the recent rule changes, we consider it highly relevant to conduct this study by stratifying data not only by weight category but also by visual category, to determine the scoring action pattern in J1 and J2 judokas.

Finally, another limitation of this study is that we did not analyze how judokas evade attacks, which often trigger counterattacks and re-counterattacks that frequently result in waza-ari. Future research that incorporates these counterattack patterns could provide a more detailed understanding of scoring dynamics in judo for visually impaired judokas.

## 5. Conclusions

This study has identified key technical–tactical patterns in visually impaired judokas, differentiated by weight categories. The results highlight the importance of adapting techniques and strategies to the specific characteristics of these athletes, revealing consistent patterns across categories.

The findings show that scoring actions concentrate in the first few minutes of the match, emphasizing the importance of early initiative. Additionally, lighter judokas tend to use counterattacks and sacrifice techniques, while heavier judokas favor direct attacks and leg counterattacks. This difference reinforces the need for training tailored to each weight category.

Visually impaired judokas effectively adapt their techniques, using grip and transitions from standing to the ground as key tools to achieve victory. This study demonstrates that these judokas not only maintain a high competitive level but also optimize their sports performance through the use of highly effective techniques.

These conclusions emphasize the importance of a personalized training approach that maximizes the tactical and technical strengths of visually impaired judokas, providing a solid foundation for the development of training programs that enhance their performance in international competitions.

**Supplementary Materials:** The following supporting information can be downloaded at: https://www.mdpi.com/article/10.3390/app142210594/s1, Table S1: Frequency, % and chi-square of the categories related to scoring actions in ground judo; Table S2: T-Patterns of scoring actions without grouped technique (individual techniques); Figure S1: Direct attack pattern in standing position using the technique Ouchi Gari in −60 kg; Figure S2: Direct attack pattern using seoi-nage in −73 kg.

**Author Contributions:** Conceptualization, I.P.-L., A.F.-M., X.R.-L.-d.-l.-O. and A.G.-S.; methodology, I.P.-L., J.C.A.-G., X.R.-L.-d.-l.-O. and A.G.-S.; software, I.P.-L. and A.G.-S.; validation, J.C.A.-G. and

X.R.-L.-d.-l.-O.; formal analysis, I.P.-L. and A.G.-S.; investigation, I.P.-L., A.J.S.-P., A.F.-M., X.R.-L.-d.-l.-O. and A.G.-S.; resources, I.P.-L., A.F.-M., A.J.S.-P. and X.R.-L.-d.-l.-O.; data curation, I.P.-L., A.F.-M., A.J.S.-P. and A.G.-S.; writing—original draft, I.P.-L., A.F.-M. and A.G.-S.; writing—review and editing, I.P.-L., X.R.-L.-d.-l.-O. and A.G.-S.; visualization, A.F.-M. and A.J.S.-P.; supervision, I.P.-L., A.F.-M. and A.G.-S.; project administration, I.P.-L., A.J.S.-P., X.R.-L.-d.-l.-O. and A.G.-S.; funding acquisition, I.P.-L. and A.G.-S. All authors have read and agreed to the published version of the manuscript.

**Funding:** This study was funded by the Ministerio de Cultura y Deporte (https://www.educacionfpydeportes.gob.es/portada.html (accessed on 20 June 2024)), Consejo Superior de Deportes (https://www.csd.gob.es/es (accessed on 20 June 2024)), and European Union (https://european-union.europa.eu/index_es (accessed on 20 June 2024)) under Project "Integración entre datos observacionales y datos provenientes de sensores externos: Evolución del software LINCE PLUS y desarrollo de la aplicación móvil para la optimización del deporte y la actividad física beneficiosa para la salud (2023)" EXP_74847 to AGS and IPL.

**Institutional Review Board Statement:** The study was approved by the ethics committee of the Faculty of Education and Sport Science (University of Vigo, application 01/1019, 15 October 2019).

**Informed Consent Statement:** Not applicable.

**Data Availability Statement:** The raw data supporting the conclusions of this article will be made available by the authors on request.

**Acknowledgments:** This publication was made possible thanks to the research stays during the years 2023 and 2024 at the Instituto Politécnico de Viana do Castelo [IPVC]—Escola Superior de Desporto e Lazer.

**Conflicts of Interest:** The authors declare no conflicts of interest.

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
