# Peer review of "The Technical–Tactical Patterns of Scoring Actions in Male Visually Impaired Judokas: A Weight Category Analysis"

_applsci, doi:10.3390/app142210594_

Round 1

Reviewer 1 Report

Comments and Suggestions for Authors

The article entitled: The technical-tactical patterns of scoring actions in male visually impaired judokas: A weight category analysis has taken into consideration the Para-judo Olympic combat. The para-Olympic sports are not as propagated as classical ones and less analysed. Sports have been shown to decrease social exclusion and reduce other accompanied body dysfunction. In the article, the authors discussed the style, strategy and effectiveness of combat observed during challenges. Moreover, they correctly selected the investigated group according to weight category. The performed statistical analysis is correct. Moreover, an etic commission agreement is not necessary for these studies (in fact open-source data). The article is well-written and readable, and the used references are correct.

In conclusion: the article is valuable for sportsmen and their coaches. Additionally, the analysis of Para-judo can improve the knowledge about judo and show their beauty without barriers between people.

Unfortunately, I recommend this article to publish in more specific MDPI journal i.e.: Sports, due to its main scope of article.

Reviewer 2 Report

Comments and Suggestions for Authors

The abstract should include more qualitative results highlighting the advantages of the outstanding studied methods.

The contributions of this paper must be included in a list in Section 1. A Section focused on Related Works must be added to the manuscript. Table 1 include a significant quantity of undefined acronyms.

Section 2 must include the evaluation metrics used to assess the studied models and systems. Tables discussion is shallow, please, add more details to better present such results.

The conclusions should be supported by the experimental results.

Comments on the Quality of English Language

The manuscript requires deep proofreading.

Reviewer 3 Report

Comments and Suggestions for Authors

Dear authors,

General Feedback 

This study explores an essential, yet understudied aspect of Para-judo: the technical-tactical patterns that lead to successful scoring actions. Given the lack of prior research on this subject, the article is an important contribution to understanding the dynamics of Para-judo for athletes with visual impairments. I have worked on a similar study and can appreciate the work that has gone into analysing the data. Well done/ 

Method Section

This section is explained in detail (i.e., even type of video recorder used) and supported by cited evidence to confirm the valid and reliable approaches used. This section also excels in its methodological rigor, employing comprehensive analytical techniques such as descriptive analysis, chi-square tests, and T-pattern analysis.  

Result Section

This section aligns with the described analysis and is conveyed in a digestible manner despite the rigorous type of analysis used. 

Please re-write this oversight "Table 2 presents the descriptive analysis of the studied sample. 179 The 172 judokas with visual impairment in the sample participated in a total of 232 180" so that it aligns with the whole paragraph. 

Discussion Section

The discussion is in-depth and refers to other studies for comparison and contrast. I do ask that the conclusion section be rewritten.  The conclusion is more or less dot points in sentence format and needs to be of a quality that mirrors the other sections. 

Kindest regards

Reviewer 4 Report

Comments and Suggestions for Authors

Dear Authors

You have written a very interesting paper focusing on identifying technical-tactical patterns of actions that result in scoring in judokas with visual impairment.

I have to commend the authors for conducting the analysis on the weight category level, as there is a great lack of studies done this way in judo overall!!! Well done.

However, some parts need to be addressed for greater clarity.

Introduction

Line 48 - and what is that particular context of para judo? Please explain it briefly to a reader who may not be from sports (have that in mind).

Lines 60-68 - you discuss the number of attacks; what about the directions of attacks and execution of techniques bilaterally? Please expand this section accordingly.

In the introduction, there is no rationale for why you chose to analyze the weight category levels. Please add. Perhaps this paper might be useful in backing up your rationale (https://www.mdpi.com/1660-4601/19/1/604)

Line 81 - as you presented it, para judokas performed techniques more efficiently and better than ''normal'' judokas. However, be critical about it. From a practical point of view, the para-judokas don't try to impose more defensive actions; therefore, we have more ''clean'' throws. On the other hand, the judokas try to escape the attack, and we have more counter-end re-counter attacks that later finish in a wazari. Also, this is a limitation of your study as your observational instrument does not include this. Please add this to the limitations. This might also be a good post-doc study/analysis of your data - a suggestion :)

Methods

Line 130 - define experts; how did you calculate their agreement and report their inter and intra-rater reliability?!

In the methods, please specify techniques, penalties, and scores—for us judokas, this is common knowledge, but not for readers from other sports or fields. Keep that in mind.

Otherwise, the statistical part is well-defined.

Results are well presented despite a large number of variables.

The discussion could be better written as it is mainly a repetition of results. Please try to connect this to the existing literature on regular judokas and add your expert knowledge on why these results might be this way - what impacted them.

The conclusion is poorly written—it should have a short summary. I suggest adding a subtitle, "Practical Application," before the Limitations section.

Overall, it is an extensive paper on an interesting population that still needs some more work.

Round 2

Reviewer 1 Report

Comments and Suggestions for Authors

The article after revision (answer to reviewers question) can be accepted for publication. However, I recommend to shift Table 4 and 6 to supplementary materials.

Author Response

Thank you very much for taking the time to review this manuscript. Please find the detailed responses below and the corresponding revisions/corrections highlighted in the re-submitted files.

Comment 1 (Comments and Suggestions for Authors)

The article after revision (answer to reviewers question) can be accepted for publication. However, I recommend to shift Table 4 and 6 to supplementary materials.

Answer Comment 1

Thank you very much for your recommendation. We will follow your suggestion and move Tables 4 and 6 to the supplementary materials. We appreciate your feedback and are confident that this change will enhance the clarity and flow of the main text.

Reviewer 2 Report

Comments and Suggestions for Authors

The authors have addressed all my previous remarks correctly.

Author Response

Thank you very much for taking the time to review this manuscript. 

Reviewer 4 Report

Comments and Suggestions for Authors

Dear Authors,

Thank you for addressing all of my comments and suggestions. The quality of your paper has improved, and it is to be accepted in its current form.

Congratulations and keep up the good work!!!

Kind regards

Author Response

(The authors gave the same response as above.)
